# Multidose Dalbavancin Population Pharmacokinetic Analysis for Prolonged Target Attainment in Patients Requiring Long-Term Treatment

**DOI:** 10.3390/antibiotics14020190

**Published:** 2025-02-13

**Authors:** Giammarco Baiardi, Michela Cameran Caviglia, Silvia Boni, Antonello Di Paolo, Valeria Marini, Giuliana Cangemi, Alessia Cafaro, Emanuele Pontali, Francesca Mattioli

**Affiliations:** 1Pharmacology and Toxicology Unit, Department of Internal Medicine, University of Genoa, 16132 Genoa, Italy; giammarco.baiardi@edu.unige.it (G.B.); michela.camerancaviglia@outlook.com (M.C.C.); valeria.marini@unige.it (V.M.); francesca.mattioli@unige.it (F.M.); 2Department of Infectious Diseases, Ente Ospedaliero Ospedali Galliera, 16128 Genoa, Italy; silvia.boni@galliera.it; 3Department of Clinical and Experimental Medicine, University of Pisa, 56126 Pisa, Italy; antonello.dipaolo@unipi.it; 4Clinical Pharmacology Unit, Ente Ospedaliero Ospedali Galliera, 16128 Genoa, Italy; 5Biochemistry, Pharmacology and Newborn Screening Unit, Central Laboratory of Analysis, IRCCS Istituto Giannina, Gaslini, 16147 Genova, Italy; giulianacangemi@gaslini.org (G.C.); alessiacafaro@gaslini.org (A.C.)

**Keywords:** dalbavancin, drug monitoring, population pharmacokinetics, PK/PD target, personalized medicine

## Abstract

**Introduction:** Dalbavancin (DAL) is a long-acting lipoglycopeptide active against Gram-positive bacteria, including multidrug-resistant isolates. A growing body of evidence supports its efficacy in various difficult-to-treat infections. DAL shows time-dependent bactericidal activity *in vitro* at free drug concentrations equal to 4×MIC values. However, the optimal dosing scheme for achieving the PK/PD target in multidose treatment has not been fully established. **Methods:** Pharmacokinetic analysis was based on a nonlinear mixed effects modelling approach performed in NONMEM v7.5/Pirana, while R was used for data management and graphical summaries. Final model parameters were used to simulate the plasma disposition of DAL by Monte Carlo simulations to determine the multidose DAL regimen associated with a 90% target attainment of 100% *f*T > 4×MIC. **Results:** A two-compartmental model with first-order elimination and allometric-scaled bodyweight best described DAL disposition in patients with CLcr > 30 mL/min. Monte Carlo simulations showed that two 1500 mg DAL doses 7 days apart granted an optimal PTA > 90% of 100% *f*T > 4×MIC up to 5, 4, and 3 weeks in patients weighting from 40–80 kg, 80–120 kg and 120–200 kg, respectively. An additional third 1500 mg dose at the above time points by weight bands may extend the optimal PTA up to 9, 7, and 6 weeks of total treatment. **Conclusions:** Two 1500 mg DAL doses administered 7 days apart could be a valuable starting strategy for patients of all weight classes with CLcr > 30 mL/min. In patients requiring long-term DAL treatment, the optimal timing of additional administrations should be guided by routine TDM or empirically through patients’ total body weight when TDM is unavailable.

## 1. Introduction

Dalbavancin (DAL) is a long-acting lipoglycopeptide active against Gram-positive pathogens, including multidrug-resistant isolates [1], approved by the Food and Drug Administration (FDA) and the European Medicines Agency (EMA) for the treatment of Acute Bacterial Skin and Skin Structure Infections (ABSSSIs) with a dosing regimen of 1500 mg as a single intravenous (IV) infusion (lasting 30 min) or 1000 mg followed by 500 mg one week apart in adults [2,3].

Noteworthy DAL structural chemical features include a polycyclic peptide core, an acylated tertiary dimethylamine side chain with enhanced activity against staphylococci, and a long lipophilic tail to facilitate dimerization and cell membrane interaction as well as to increase the half-life of the drug. DAL exerts its mechanism of action by disrupting cell wall synthesis in Gram-positive bacteria, through the formation of a stable complex between the drug and the d-alanyl-d-alanyl portion of the bacterial wall; it also possesses increased binding affinity to the target site through the lipophilic side chain resulting in higher potency than other glycopeptide antibiotics, allowing anchoring to cell membranes and thus enabling longer duration of action [4].

ABSSSIs represent some of the most frequently treated infections in hospital settings. Currently, ABSSSIs remain the only approved indication for DAL, although this may be a limitation to its use by not taking full advantage of its pharmacokinetic/pharmacodynamic (PK/PD) properties.

DAL’s peculiar PK/PD properties, consisting of a long terminal half-life (about 14.4 days), high binding to plasma proteins (93%), predominant non-renal clearance (CL), and good tissue penetration, allow indeed for long-term efficacy [5].

Moreover, DAL clearance (CL) does not appear to be affected by the presence of cytochrome p450 substrates, inhibitors, or inducers, or the presence of concomitant drugs. In addition, age, sex, and ethnicity of the patient do not seem to have any impact on its PK profile [6]. Conversely weight and renal function may have an impact on its pharmacokinetics [7].

However, no dose adjustments of DAL are required in mild-to-moderate renal failure, any degree of liver failure, and renal replacement therapy modality (e.g., intermittent hemodialysis, peritoneal dialysis) [8,9]. Dose reduction should instead be implemented only in patients with severe renal impairment (creatinine clearance—CLcr < 30 mL/min) but not in patients receiving renal replacement therapy.

The ratio of area under the plasma free-drug concentration–time curve (*f*AUC) to minimum inhibitory concentration (*f*AUC/MIC) appears to be the PK/PD parameter that correlates with the *in vivo* efficacy of DAL [10,11]. However, the time the concentration of free DAL remains above the MIC (*f*T > MIC) was believed to better predict treatment results, since DAL shows time-dependent bactericidal activity *in vitro* against staphylococci and streptococci at free drug concentrations equal to 4×MIC values [12,13] and a very long elimination half-life.

It has been suggested that a trough plasma concentration (C_trough_) of DAL > 8.04 mg/L is associated with a high probability of target attainment against *S. aureus* with an MIC up to the former EUCAST clinical breakpoint of susceptibility for DAL (0.125 mg/L) [14].

A growing body of evidence supports the efficacy of DAL as a long-term therapy in infections such as osteomyelitis, prosthetic joint infections, endocarditis, central venous catheter-associated bacteremia, pediatric infections, and infections in diabetic patients [15,16], in which a treatment of at least 6 weeks is usually required [17,18]. Because of its PK/PD properties, DAL could thus be a viable alternative to standard regimens based on daily antibiotic intravenous infusions for outpatients requiring a long-term treatment of Gram-positive bacterial infections.

Thus, in addition to the previous label schedule other off-label schedules have been proposed for different indications beyond ABSSSIs [19]; however, clinical evidence has shown wide heterogeneity in DAL dosing schedule and treatment duration in different real-world clinical scenarios.

Dunne M. et al. [4] suggest that only two doses of DAL (1500 mg on day 1 and 1500 mg on day 8) can provide sufficient tissue exposure in *Staphylococcus aureus* infections for 8 weeks. A recent expert opinion [20] suggests the use of Therapeutic Drug Monitoring (TDM) in guiding the timing of the subsequent dose of DAL (beyond the total dose of 3000 mg), to detect underdosing, reduce the risk of emergence of resistant bacterial strains, and thus reduce costs to healthcare systems, too.

However, to date, there is no rationale for the standardization of the treatment approach in patients requiring long-term (>6 weeks) DAL treatment.

Thus, this study aims to characterize the population pharmacokinetics of multidose DAL and to optimize DAL prolonged (>6 weeks) antibiotic therapy maximizing its efficacy in the treatment of difficult infections and reducing the spread of resistant bacteria according to PK/PD and antimicrobial stewardship principles.

## 2. Results

### 2.1. Study Population

A total of 30 adult patients were included in the present study. The clinical and demographic characteristics of the population are presented in Table 1. Overall, each patient received a multidose DAL regimen that included at least two 1500 mg DAL administrations 7 to 14 days apart at the discretion of the prescribing physician. Each patient was monitored at C_trough_ and C_max_ levels at each DAL administration with additional sparse sampling whenever feasible. The number of DAL doses administered and TDM instances per patient is reported in Table 1.

### 2.2. Dalbavancin Population Pharmacokinetics

A total of 195 DAL concentration values were initially included in the population pharmacokinetic analysis (popPK).

A two-compartmental model with first-order elimination best described the DAL PK. Body weight scaled by allometric principles [21,22,23] was set *a priori* as a covariate with fixed exponents of 0.75 on clearances (CL, Q) and of 1 on volumes of distribution (V1, V2).

Other possible covariates with a mechanistic and physiologic rationale failed to improve the fitting of the observed data into the classical diagnostic plots or the model performance in terms of ΔOFV. Interestingly, the addition of the effect of CLcr on CL by means of linear and non-linear relationships failed to improve the model performance in terms of differences in objective function value (ΔOFV; 0.02), although in other popPK models adding the covariate explained part of the inter-individual variability (IIV) on CL [24]. This may be mechanistically explained by the DAL renal elimination pathway that involves only one-third of a dose to be excreted unchanged in the urine. Consequently, dose adjustments are required only for patients with severe renal impairment (CLcr < 30 mL/min). Conversely, in our patient population, CLcr was always >30 mL/min (Table 1).

During model building, five PK data points were identified as outliers (CWRES > 3) and were thus removed from the dataset. Interestingly, upon removal the model achieved a significant improvement in terms of ΔOFV (−100.834).

Final population parameters estimates presented in Table 2 show good reliability (RSE% < 30%, eta shrinkage < 35%) which is further sustained by bootstrap analysis, and by the visual assessment of graphical key diagnostics plots (Figure 1) and pcVPC (Figure 2).

### 2.3. Pharmacodynamic Target Attainment of Dalbavancin Long-Term Use Dosing Strategy

Monte Carlo (MC) simulations of 100,000 DAL profiles (see Section 4 were performed from a virtual patients’ population weighing from 40 to 200 Kg given the identified strong mechanistic and physiological covariate effect of weight on DAL PK parameters. An initial two-dose regimen consisting of a first administration of 1500 mg at day 0 followed by1500 mg at day 7 was tested on the overall population to select the optimal timing of next dose administration that returned a probability of target attainment (PTA) ≥ 90% for each weight band. Multiple MC simulations showed that two 1500 mg DAL doses 7 days apart granted an optimal PTA > 90% of 100% *f*T > 4×MIC (MIC set at 0.25 mg/L, EUCAST/USCAST clinical breakpoint) up to 5, 4, and 3 weeks in patients weighting from 40–80 kg, 80–120 kg, and 120–200 kg, respectively (Figure 3). An additional third 1500 mg dose at the said respective time points by weight bands extended the optimal PTA to 9, 7, and 6 total weeks of treatment (Figure 4).

The simulated free DAL concentration profiles over time related to the initial DAL long-treatment regimen (1500 mg at day 0 + 1500 mg at day 7) and personalized subsequent administrations by weight bands are represented in Figure 5.

## 3. Discussion

This popPK study was conducted in a cohort of adult patients affected by difficult-to-treat infections who needed long-term antimicrobial therapy irrespective of the site of infection.

The popPK of dalbavancin in adults was investigated during two pivotal clinical trials involving ABSSSI patients [6,25] and in real-life patients with osteoarticular infections [26], ventricular assist device infections [27], and subacute/chronic infections [24]. Most of these studies used a two-compartment model as does our own; however, the study of Cojutti et al. (2022) [24], in which the clinical characteristics of the studied population best align with ours, reports higher population distributive volumes values (V1 and V2) probably due to the inability to appropriately estimate the inter-individual variability in both pharmacokinetic parameters (RSE% > 30). Interestingly, in our study, adding the covariate effect of CLcr on CL did not benefit the model performance [6,24,25].

This may be mechanistically explained by the DAL elimination pathway which occurs via a combination of renal (approximately 45%) and nonrenal clearance [28], suggesting that nonrenal pathways, or further elimination pathways not fully characterized, play a major role in dalbavancin elimination [29]. Consequently, DAL dose adjustments are required for patients with severe renal impairment (CLcr < 30 mL/min) [2]. Conversely, in our patient population, CLcr was always >30 mL/min, suggesting a major role of total body weight as covariate effect on the pharmacokinetic parameters of DAL in this clinical context.

Currently, DAL multidose treatment regimens are increasingly employed in real-world clinical practice as a repeated administration of a second dose 7 days after the first, with further doses at timings for which a clear agreement has not yet been reached. Several difficult-to-treat conditions may benefit from three or more DAL administrations, such as osteomyelitis, spondylodiscitis, septic arthritis, prosthetic joint infections, endocarditis, central venous catheter-associated bacteremia, and diabetic foot infections [15,16,17,18,30,31,32,33,34,35]. In some instances, TDM has been proposed to help personalizing the frequency of DAL re-administration in difficult-to-treat infections [34,35,36] since DAL exhibits potent activity *in vitro* against established biofilms due to *Staphylococcus aureus*, *Staphylococcus epidermidis*, *Enterococcus* spp. vancomycin-sensitive (VSE), or vancomycin-resistant Van B [37,38] thus playing a crucial role in the management of relevant infections characterized by bacterial biofilm production.

DAL showed time-dependent bactericidal activity *in vitro* at free drug concentrations equal to 4×MIC values [12,13]; however, the definition of the clinical DAL PK/PD target varies from one study to another [14,36,39]. The initial *in vivo* PK/PD index of *f*AUC/MIC derived for bacterial stasis, based on a neutropenic murine thigh model, has been substantially revised in the years since the initial publication (*f*AUC/MIC > 265) [10] with a change of approximately 10-fold in a later publication (*f*AUC/MIC > 27.1) [11].

Moreover, earlier *in vitro* studies [12] were not taken into account when first testing dalbavancin PK/PD targets *in vivo* [10].

For these reasons, and the very long DAL t_1/2_, our PTA simulations were based on a PK/PD index of 100% *f*T > 4×MICas surrogate marker of efficacy in order to predict the best timing of DAL next-dose administration which would guarantee that the overall population (≥90%) would not fail this specified PK/PD target over time.

From our results, a second 1500 mg DAL dose 7 days apart from the first (day 0) guarantees an optimal (>90%) PTA of 100% *f*T > 4×MIC (MIC set at 0.25 mg/L, EUCAST/USCAST clinical breakpoint) up to 5, 4, and 3 weeks in patients weighting from 40–80 kg, 80–120 kg, and 120–200 kg, respectively. This evidence suggests monitoring carefully the clinical response of overweight patients when treated with multidose DAL regimes for difficult infections, because they may fail the PK/PD target earlier than expected. An additional third 1500 mg dose at the said respective time points by weight bands may further extend the optimal PTA to 9, 7, and 6 total weeks of treatment, which would fit within the recommended time length for the majority of difficult-to-treat Gram-positive infections.

The evidence of a *f*T > 4×MIC with a MIC set at 0.25 mg/L (EUCAST/USCAST clinical breakpoint) [40,41] represents a conservative estimate for target attainment which may not be required for efficacy purposes against all isolates below this MIC breakpoint. However, targeting the highest reported MIC clinical breakpoint/epidemiological cut-off may be a potential strategy against bacterial resistant strain dissemination [42,43,44].

Consequently, TDM may also play a role in personalizing the timing of DAL next-dose administration in association with microbiological evaluation of the MIC of the bacterial isolate to target a *f*DAL concentration > 4×MIC. In the absence of clear culture results, we suggest maintaining at least a total DAL concentration > 14.29 mg/L, which corresponds to a *f*DAL concentration > 4×MIC (MIC set at 0.25 mg/L, EUCAST/USCAST clinical breakpoint) for resistant strain diffusion prevention.

The calculation method for obtaining the total DAL concentration value corresponding to a *f*DAL concentration > 4×MIC when assuming a 7% free fraction of DAL in plasma is the following:fDAL>4×MICIf fDAL=0.07×TotalDAL and MIC=0.25 mg/L$$TotalDAL>\frac{4\times 0.25\;mg/L}{0.07}$$TotalDAL>14.29 mg/L

Recently, initial TDM targets for total DAL concentration over time (namely 4.02 mg/L for the attainment a *f*AUC 24/MIC ratio > 111.1, against staphylococci with a MIC up to 0.0625 mg/L, or 8.04 mg/L for a MIC up to 0.125 mg/L) [14] have been revised to a single TDM target of total DAL > 14.5 mg/L at any time point which may achieve C-reactive protein production inhibition over time in >95% of the treated patients with osteoarticular infections [45]. This efficacy TDM target is consistent with our hypothesis that a *f*DAL concentration > 4×MIC (total DAL > 14.29 mg/L) should be maintained over time for resistant strain diffusion prevention and further evaluated during efficacy studies to establish a clear DAL PK/PD target via exposure–response analysis of clinical data.

## 4. Materials and Methods

### 4.1. Study Design

This prospective/retrospective study was approved by Liguria Territorial Ethics Committee (DALT DRUM Study, Registration number 82/2023—DB id 13014) and carried out from February 2023 to February 2024 at Ente Ospedaliero Ospedali Galliera, Genoa, Italy. Adult patients referred to the Department of Infectious Diseases with documented or suspected Gram-positive infection who failed primary antimicrobial therapy and were prescribed a multidose DAL regimen as a rescue treatment were deemed eligible for inclusion and after signing the informed consent they were enrolled in the study.

Patients were started on a 1500 mg DAL dose. Subsequent individual DAL dose amount and timing of drug administration were established by the infectious disease consultant based on the clinical evaluation of infection severity and/or progression, the sensitivity of bacterial strain, and on the trends of inflammatory biomarkers.

Blood samples for PK analysis were collected during routine visits immediately before the administration of DAL and at the end of infusion, to monitor C_trough_ and C_max_ levels, respectively. To avoid impacting patient lifestyle and routine clinical practice, a sparse sampling strategy was also adopted for outpatients needing blood tests at the convenience of the prescribing physician. This sampling strategy was intended to reduce the risk of drug exposure overestimation and capture the overall DAL disposition.

DAL total plasma concentrations were quantitated by means of the liquid chromatography-tandem mass spectrometry method previously described by Cafaro et al. [46] and validated in accordance with the most recent ICH guidelines M10 on bioanalytical method validation [47], in the calibration range 0.66–800 mg/L.

The actual time of each DAL infusion duration and PK sample collection was recorded in the source document along with patient demographic (age, gender, weight, height), clinical (type and site of infection, microbiological isolates), and laboratory (serum creatinine, creatinine clearance by the Cockcroft–Gault equation, serum albumin, C-reactive protein) data.

### 4.2. Population Pharmacokinetic Modeling

Pharmacokinetic data were analyzed according to a nonlinear mixed-effects modeling approach using NONMEM v7.5 with Pirana modeling workbench [48], whilst R, version 4.3.2, was used for data formatting and graphical and statistical summaries. No data transformation was applied to PK concentrations prior to the analysis.

From an initial model (one-compartment, first-order elimination with proportional error model), different possible combinations of structural and stochastic models, as well as the inter-individual variability (IIV) of PK parameters were assessed. The effect of several covariates, previously screened for their influence on the PK parameters of the drug under study, was then included in a stepwise manner with backward elimination in the final model. Continuous covariates (age, weight, height, serum albumin, creatinine, creatine clearance, PCR) were centered on their median value and their effect was evaluated by linear and non-linear relationships. Improvement between models was judged based on a decrease in objective function value (OFV) greater than 3.84 units (*p* < 0.05) using the FOCE-I estimation method, whilst an increase by more than 6.64 units (*p* < 0.01) upon removal was adopted during backward exclusion. Additionally, during the initial model-building process outliers were detected and excluded from the analysis based on visual examination of data points identified with conditional weighted residuals (CWRES) > 3. To evaluate the influence of outliers, a sensitivity analysis was conducted in which the updated popPK model was run with and without outliers. If a statistically significant change in OFV or in the estimation of key fixed-effects parameters was observed, outliers were considered influential in the modeling steps and commented out [49].

### 4.3. Model Evaluation

The predictive performance of the PK models was assessed by appropriateness of the relative standard error (RSE < 30%), graphically by means of classical key diagnostic plots [49], prediction-corrected visual predictive check (pcVPC) [50], and by inspecting bootstrap results from 1000 resampled datasets. Finally, eta-shrinkage values were calculated to identify and quantify model overfitting.

### 4.4. Monte Carlo Simulations to Establish the Dose Rationale of Dalbavancin Long-Term Treatment

The final model was used to simulate DAL plasma disposition in a virtual population consisting of 3000 adults for whom covariate data were harvested from real published datasets [51], so that the relationship between each covariate would be physiologically based.

The virtual population was generated with R and consisted of three weight-banded groups of individuals (1000 each) weighing from 40 to 200 kg, which was deemed sufficient to evaluate the effect of covariates and stochastic variability on systemic exposure of DAL. Monte Carlo simulations from the final PK model were performed in NONMEM v7.5 to generate 100,000 DAL PK profiles.

Different dosing regimens of DAL were tested to evaluate the dose rationale for a long-term treatment in patients affected by difficult-to-treat infections. A DAL regimen of 1500 mg at day 0 + 1500 mg at day 7 was chosen as the best initial regimen when expecting to treat the patient for more than 6 weeks based on Senneville E. et al. [20]’s expert review panel, ease of administration, patients’ convenience, and control over the patient’s clinical course. Timing of subsequent DAL administrations was calculated based on PTA ≥ 90% of the PK/PD index of 100% *f*T > 4×MIC, which was deemed an adequate surrogate marker of efficacy. 

The PK/PD target of 100% *f*T > 4×MIC was set by extrapolating DAL time-dependent bactericidal activity from *in vitro* studies at free drug concentrations equal to 4×MIC values [12,13], and the MIC distribution was reported based on EUCAST MIC value distribution [52]. The MIC cut-off value for the analysis was set at 0.25 mg/L (EUCAST/USCAST clinical breakpoint) [40,41].

DAL shows high binding to plasma proteins (93%) [28]; however, the calculated free drug concentrations of DAL in plasma were very similar to the drug levels measured in bone by Dunne M. et al. [4]. Thus, consistent with this finding and as reported on the label [2], we considered the free fraction of DAL in plasma (7%) to be freely available for antimicrobial activity over time during a long-term treatment, even to poorly accessible sites of infection [53,54].

## 5. Conclusions

Our study strongly suggests personalizing the dosing schedule of DAL according to individual total body weight of each patient, hence offering the best systemic exposure for a time interval required to treat severe infections.

In particular, in the absence of a TDM service, during prolonged DAL multidose treatment the timing of subsequent DAL administration may be guided empirically by individual total body weight for patients with CLcr > 30 mL/min. Optimal timing of additional administrations should be guided by microbiological MIC isolate identification to target *f*DAL concentration > 4×MIC over time (total DAL concentration > 14.29 mg/L for a MIC up to the updated clinical breakpoint).

## Figures and Tables

**Figure 1 antibiotics-14-00190-f001:**
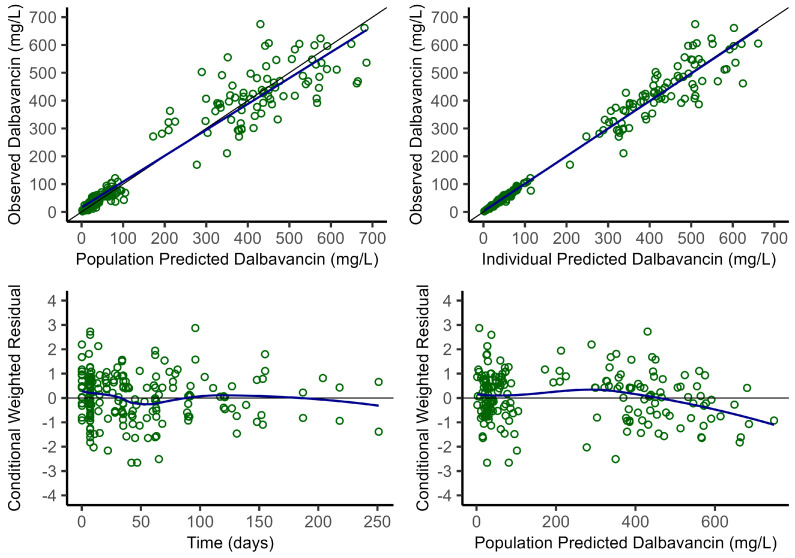
Goodness of fit key classical diagnostic plots.

**Figure 2 antibiotics-14-00190-f002:**
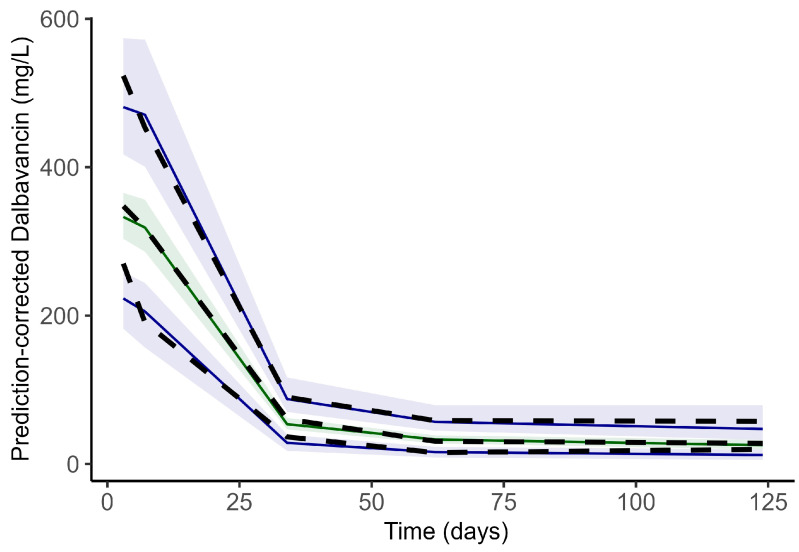
Prediction-corrected visual predictive check. Dashed lines: fifth, 50th, 95th percentiles of observed data. Colored areas: 95% confidence interval of the fifth, 50th, 95th percentiles of simulated data with their respective medians (solid lines).

**Figure 3 antibiotics-14-00190-f003:**
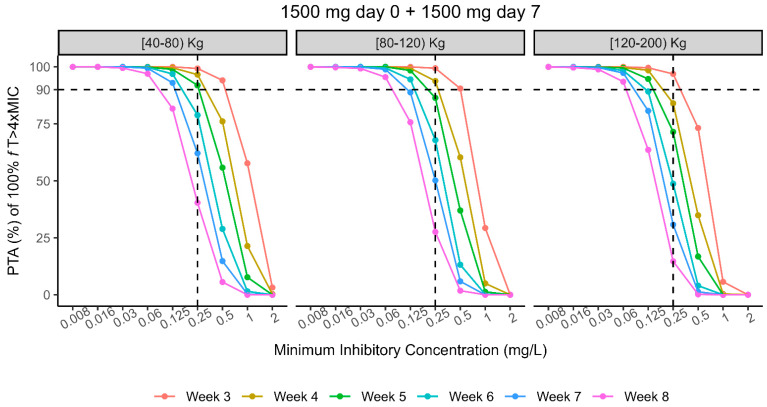
Percentage of probability of target attainment (PTA%) of DAL 1500 mg on day 0 and day 7, by classes of weight.

**Figure 4 antibiotics-14-00190-f004:**
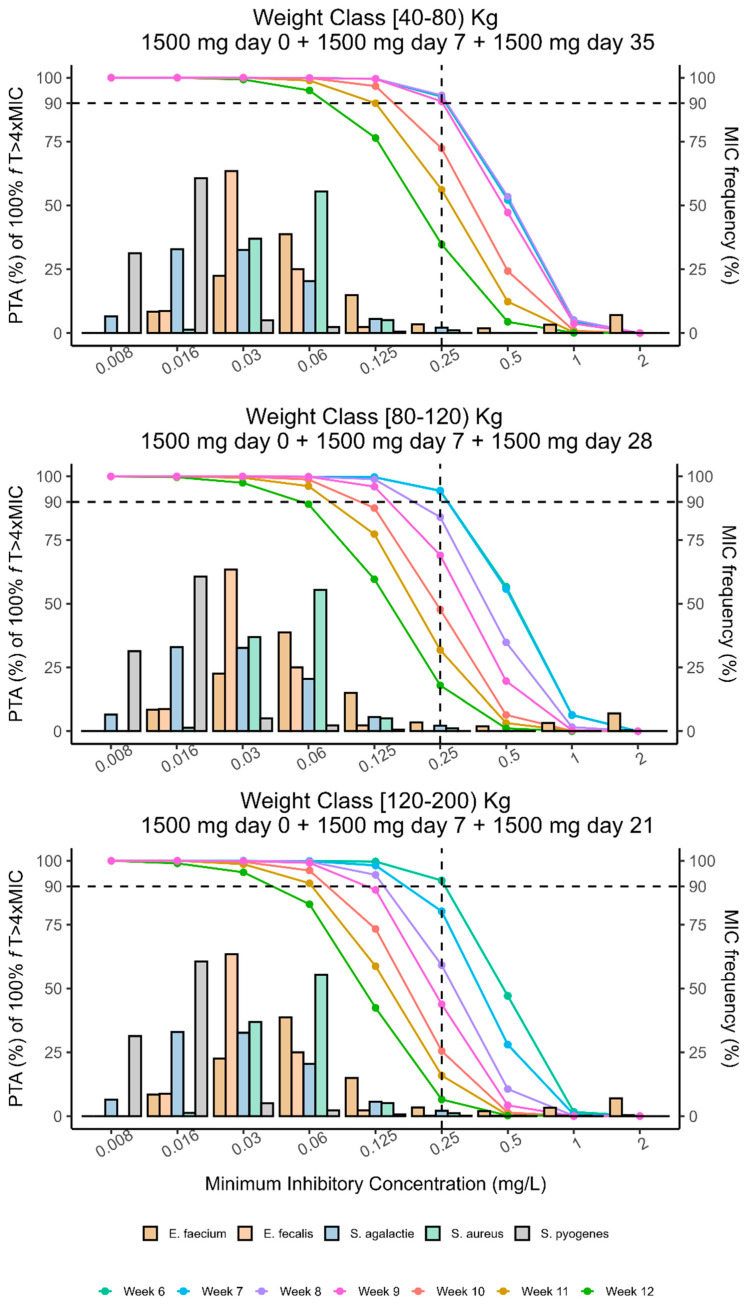
Percentage of probability oftarget attainment (PTA%) of DAL personalized multidose regimen by classes of weight over EUCAST MIC frequency distribution.

**Figure 5 antibiotics-14-00190-f005:**
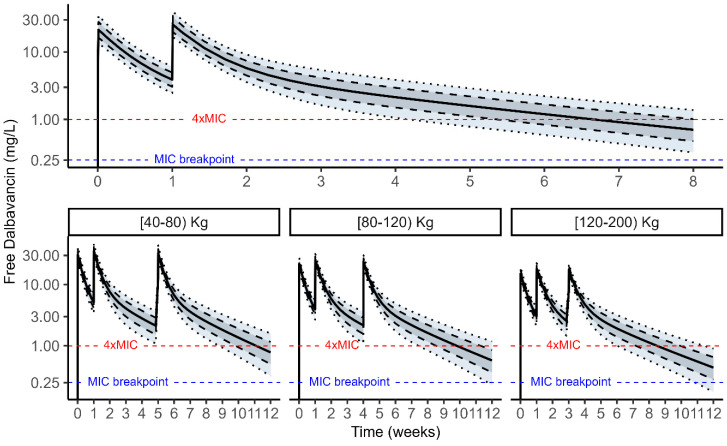
Simulated free dalbavancin plasma concentration versus time profiles of an initial two 1500 mg dosing regimen administered one week apart in the overall population (**top** panel) and of personalized timing of subsequent administrations in different classes of weight (**bottom** panels) chosen by PTA analysis. The solid line is the simulated median concentration vs. time profile. The dotted and dashed lines are the 10th and 90th and 25th and 75th percentiles of simulated concentrations vs. time, respectively. The horizontal dashed blue line represents the EUCAST/USCAST clinical breakpoint MIC value of 0.25 mg/L, while the horizontal dashed red line corresponds to 4×MIC.

**Table 1 antibiotics-14-00190-t001:** Study population clinical and demographic characteristics (n = 30).

Characteristics	Value *
Gender (n. male/n. female)	21/9
Age (years)	72 (26–97)
Height (cm)	168 (140–195)
Weight (Kg)	72 (44–179)
Body Mass Index (Kg/m^2^)	25.8 (17.3–81.6)
Creatinine (mg/dL)	1 (0.5–1.7)
Creatinine Clearence Cockcroft–Gault (mL/min)	66.2 (31.7–283)
C-Reactive Protein (mg/L)	8.4 (0.3–183.7)
Serum Albumin (g/L)	33.1 (21.4–39.6)
Number of DAL doses per patient	3 (2–10)
Number of TDM instances per patient	5.5 (2–20)
**Isolate**	
MSSA	4
MRSE	5
MRSA	5
*S. agalactie*	1
*S. epidermidis*	1
*S. sanguinis*	1
*S. lugdunensis*	1
NG/NA	12
**Site of Infection**	
ABSSSI	10
Spondylodiscitis	9
Septic arthritis	5
Endocarditis	2
Chronic knee bursitis	1
Osteomyelitis	1
Spinal implant infection	1
Prostatic abscess	1

* Values are expressed by Median (Minimum–Maximum) for continuous variables Abbreviations: ABSSSI: Acute Bacterial Skin and Skin Structure Infection; MRSA: methicillin-resistant *Staphylococcus aureus*; MSSA: methicillin-susceptible *Staphylococcus aureus*; MRSE: methicillin-resistant *Staphylococcus epidermidis*; NG/NA: no growth or culture results not available.

**Table 2 antibiotics-14-00190-t002:** Final model parameters estimate.

Fixed Effects	Estimate	RSE (%)	Bootstrap [Median (CI 5—95%)]	
CL (L/h)	0.0273	5.1	0.0272 (0.0251–0.0294)	
V1 (L)	3.6	3.7	3.6 (3.4–3.8)	
Q (L/h)	0.0225	28.4	0.0223 (0.0175–0.02936)	
V2 (L)	6.4	11.9	6.5 (5.6–7.5)	
**Random Effects (CV%)**				**Eta Shr. (%)**
ω CL	22%	13.6	21.4% (16.0–26.3)	8
ω V1	17.3%	17.4	16.9% (11.0–21.2)	17
ω Q	55.9%	21.2	53.0% (31.3–71.1)	30
ω V2	30.1%	15.9	28.6% (19.4–37.4)	34
**Residual Variability**				
b	0.144	10	0.144 (0.126–0.166)	21

Abbreviations: RSE: Relative standard error of the estimate; b: Residual error (proportional); CL: Total body clearance; Q: Inter-compartmental clearance; V1: Central volume of distribution; V2: Peripheral volume of distribution; ω: Interindividual variability; CI: Confidence interval; Eta Shr.: Eta shrinkage.

## Data Availability

The clinical and laboratory data supporting this study’s findings are available from the corresponding authors E.P. and F.M. upon special request. The datasets generated or analyzed during the current study are not publicly available for ethical reasons per local guidelines.

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
