# Peer review of "Multidose Dalbavancin Population Pharmacokinetic Analysis for Prolonged Target Attainment in Patients Requiring Long-Term Treatment"

_antibiotics, 2025, doi:10.3390/antibiotics14020190_

Round 1

Reviewer 1 Report

Comments and Suggestions for Authors

The authors present a compelling and relevant paper exploring the pharmacokinetics of dalbavancin in repeated dosing, assessing the probability of target attainment in different populations. Given the emerging possibility of dalbavancin use in infections that traditionally require long-term frequent administration of other antibiotics, such exploration is clinically relevant and useful. 

Unfortunately, the methodology of the presented paper falls short in robustly supporting the claims made. Since we are in off-label territory, any study hoping to inform dosing recommendations needs to withstand heightened scrutiny if it is to be clinically useful. The main issue (see more detailed discussion below) I see with the presented work is the lack of free fraction measurement and lacking discussion of possible effects of altered binding on the results, should they be based solely on the 93% binding included in the drug label. It may well be reasonable, but a more thorough discussion needs to be included, and patient characteristics better explored to justify this approach. The lack of other relevant covariates besides body weight is unexpected but may reflect a possible homogenous nature of the study population (characteristics underreported by authors). Using it to extrapolate into extremes of values that were not properly represented in the study population may be unwarranted. 

In the absence of free fraction measurements, there is only so much the authors can do to address the binding issue, however a possible difference between 7 % or 1 % of free fraction is too large to ignore. The paper provides a valuable insight into the pharmacokinetics and dosing of dalbavancin but to be used as a basis for off-label prescribing, the rigour justifying the claims must be increased. A discussion of constraints on the results achieved is necessary. 

Methodological issues

1)       Albumin binding

This is a major omission. The paper reports (lines 338 and 350) that albumin levels were measured and analysed but these are not reported or discussed further at all. Given that the reported 93% binding of DAL in the drug label may not reflect the true in vivo binding, a discussion of reported higher values (up to 99 % in Turner et al, see below) is warranted as it will significantly affect the PTA values for the free fraction. Omitting reporting of albumin levels and the possible impact of different protein binding values in the patient population is a major methodological flaw that severely limits any real-life application of the conclusions that paper arrives at; this needs to be reported and discussed in more detail. 

2)       Parameter distribution in the study population

Authors only report means with maximum and minimum values without giving any other insight into the actual distribution of these values in the studied population. This is especially relevant for the only included covariate of body weight given the extrapolation used for the simulation.

3)       Extrapolating into unjustified territory

Given the lacking reporting on value distribution in the population, it is difficult to make authoritative statements, but the expected number of patients over 120 kg in the studied sample is in low single digits, making the extrapolation to 200 kg in the Monte Carlo simulation unjustified. See discussion on using TBW further. 

4)       Discussion of body weight

There is no discussion of using alternative body weight formulas such as adjBW or IBW for modelling as covariates. Given only moderate lipid solubility of dalbavancin, this could be relevant for dosing at extreme end of the distribution. While most studies on DAL use TBW as a covariate, this still needs to be discussed and explored since the paper ventures into the upper extreme of body weights where the potential bias of TBW might manifest. Given the off-label nature of these recommendations, any suggestion for dosing needs to be properly justified to avoid exposing the patient to excessive concentrations.

5)       Excluded outliers

Please include a more thorough discussion on the nature and justification of the excluded outlier values. Just saying that their removal improved the model fit is not sufficient. 

6)       Covariates in time

It is not clear from the text if covariates such as creatinine clearance or CRP were measured at repeated intervals (at sampling times, at repeated dosing) and if so, if the changing values were incorporated when building the model. Given the longer duration of follow-up (at least 7 days between doses, one patient with 10 doses) and the wide range of initial values, it would be interesting to see a discussion on this topic. 

7)       Exploring other PK/PD parameters

Given the trough levels achieved, it seems prudent to assume the simulated profiles would also satisfy the AUC:MIC PK/PD target. Even though the authors opted to use fT<4×MIC and justified their choice, it seems that a comment on how these profiles would fare in the other parameter would be interesting. 

8)       Missing references

There are several relevant papers that are missing from the discussion, namely a real world analysis of achieved levels in Hervochon et al. Journal of Antimicrobial Chemotherapy, Volume 78, Issue 12, December 2023, Pages 2919–2925, https://doi.org/10.1093/jac/dkad331 and more importantly analysis of protein binding in Turner   NA, Xu   A, Zaharoff   S  et al.   Determination of plasma protein binding of dalbavancin. J Antimicrob Chemother  2022; 77: 1899–902. https://doi.org/10.1093/jac/dkac131

9)       Covariates and Monte Carlo simulation

It is not clear what is meant on line 366 by “..so that the relationship between each covariate would be physiologically based.” when the only covariate included in the model was body weight. The paper does not mention which covariates were evaluated, only that CLCr did not improve the fit. What were the other covariates considered? One could expect albumin levels to affect free fraction and thus clearance, if there was variability in the population (values not reported by authors). 

10)   Minor issues

A minor quibble, but in Figure 1, the axes labels are inconsistent - X axes should also include units for completeness and Y axes should make it clear that those are measured levels (as technically, both axes are dalbavalcin). This may be clear to people familiar with goodness of fit plots, but for completeness the labels should be correct. 

On line 95, please consider mentioning that the cited EUCAST clinical breakpoint cited is no longer current, as the paper itself uses the new updated breakpoint of 0.25 mg/L. 

Despite the study not being designed or powered to assess outcomes, it would be interesting to see the real world MIC values in the pathogens that were identified and treatment outcomes, if available. 

Author Response

The authors present a compelling and relevant paper exploring the pharmacokinetics of dalbavancin in repeated dosing, assessing the probability of target attainment in different populations. Given the emerging possibility of dalbavancin use in infections that traditionally require long-term frequent administration of other antibiotics, such exploration is clinically relevant and useful. 

Unfortunately, the methodology of the presented paper falls short in robustly supporting the claims made. Since we are in off-label territory, any study hoping to inform dosing recommendations needs to withstand heightened scrutiny if it is to be clinically useful. The main issue (see more detailed discussion below) I see with the presented work is the lack of free fraction measurement and lacking discussion of possible effects of altered binding on the results, should they be based solely on the 93% binding included in the drug label. It may well be reasonable, but a more thorough discussion needs to be included, and patient characteristics better explored to justify this approach. The lack of other relevant covariates besides body weight is unexpected but may reflect a possible homogenous nature of the study population (characteristics underreported by authors). Using it to extrapolate into extremes of values that were not properly represented in the study population may be unwarranted. 

In the absence of free fraction measurements, there is only so much the authors can do to address the binding issue, however a possible difference between 7 % or 1 % of free fraction is too large to ignore. The paper provides a valuable insight into the pharmacokinetics and dosing of dalbavancin but to be used as a basis for off-label prescribing, the rigour justifying the claims must be increased. A discussion of constraints on the results achieved is necessary. 

A: The Authors wish to thank the Reviewer for his/her introductory assessment, which shed insight and sets the stage for a thorough reflection on some aspects that the Authors do not fully agree with and reserve the right to respond point by point as follows.

Methodological issues

1)       Albumin binding

This is a major omission. The paper reports (lines 338 and 350) that albumin levels were measured and analysed but these are not reported or discussed further at all. Given that the reported 93% binding of DAL in the drug label may not reflect the true in vivo binding, a discussion of reported higher values (up to 99 % in Turner et al, see below) is warranted as it will significantly affect the PTA values for the free fraction. Omitting reporting of albumin levels and the possible impact of different protein binding values in the patient population is a major methodological flaw that severely limits any real-life application of the conclusions that paper arrives at; this needs to be reported and discussed in more detail. 

A: The Authors recognize that there was some issue in the savings of the final version of Table 1 (also reported by Reviewer 3 for a missing value). Now missing values for albumin and bacterial isolates are amended.

However, regarding dalbavancin protein binding / free fraction measurement, it is worth mentioning how many authors consider it complex to measure the fraction of drug not bound to plasma proteins. Turner himself clearly states how complex it is to measure free dalbavancin without incurring in errors of underestimation or upper estimation of the free concentration (e.g. non-specific binding to plastic, presence or absence of Tween 80, radiochemical purity, etc.).
Probably the action of a drug changes only when protein binding changes, and it is certainly true that in the case of highly bound drugs, small changes in the bound fraction result in large changes in the unbound concentration. However, the in vitro observation that a change in protein concentration results in a change in free drug concentration does not necessarily apply to the in vivo situation.
Unless there is a sudden shift from binding by drug-drug interaction, changes are usually gradual and do not alter the effects of the drug because the compensatory increase in the rate of elimination (Clearance) from the body prevents the accumulation of a high concentration of bound drug in the plasma.
It should also be pointed out that for lipophilic drugs (i.e. dalbavancin), free drug concentration in plasma is determined primarily by equilibrium with other tissues, not by equilibrium with plasma proteins. Therefore, free drug concentration in plasma is not highly influenced by changes in protein binding for lipophilic drugs in vivo following equilibration with peripheral tissues. 

If only the unbound drug concentration was to be measured, for lipophilic drugs such as dalbavancin, there would be almost no change in the apparent volume of distribution (Vd), since the free drug concentration in plasma after equilibration with peripheral tissues is only trivially affected by the bound drug in plasma.

Finally, it should be remembered that in many diseases and other abnormal physiological states there is a change in the distribution of albumin between plasma and interstitial fluid, and this may be accompanied by a change in the corresponding fluid volumes (Faed 1981). Jusko and Gretch (1976) pointed out that measurements of plasma protein concentrations may be an insufficient index of total body levels of albumin, since patients differ considerably in the distribution of the protein between plasma and interstitial fluid. Turner himself concluded that “Future studies are also required to determine whether therapeutic effect is best predicted by free or total drug concentration as this remains unverified for dalbavancin”.

Faed EM. Protein binding of drugs in plasma, interstitial fluid and tissues: effect on pharmacokinetics. Eur J Clin Pharmacol. 1981;21(1):77-81. doi: 10.1007/BF00609592. PMID: 7333350.

Schmidt S, Gonzalez D, Derendorf H. Significance of protein binding in pharmacokinetics and pharmacodynamics. J Pharm Sci. 2010 Mar;99(3):1107-22. doi: 10.1002/jps.21916. PMID: 19852037.

Wanat K. Biological barriers, and the influence of protein binding on the passage of drugs across them. Mol Biol Rep. 2020 Apr;47(4):3221-3231. doi: 10.1007/s11033-020-05361-2. Epub 2020 Mar 5. PMID: 32140957.

Jusko WJ, Gretch M. Plasma and tissue protein binding of drugs in pharmacokinetics. Drug Metab Rev. 1976;5(1):43-140. doi: 10.3109/03602537608995839. PMID: 829788.

In conclusion measuring dalbavancin total drug concentration is recommended over the free fraction itself (as for many other drugs for which Therapeutic Drug Monitoring is warranted) and albumin should not highly affect efficacy in real world patients, or at least adults one. Despite nowadays many population pharmacokinetics models exist for dalbavancin only paediatric ones retain albumin as covariate, probably due to the large volume of distribution changes that affect pediatrics during growth processes, which are present no more in adults. Moreover, our patients show albumin levels within normal ranges.

Gonzalez D, Bradley JS, Blumer J, Yogev R, Watt KM, James LP, Palazzi DL, Bhatt-Mehta V, Sullivan JE, Zhang L, Murphy J, Ussery XT, Puttagunta S, Dunne MW, Cohen-Wolkowiez M. Dalbavancin Pharmacokinetics and Safety in Children 3 Months to 11 Years of Age. Pediatr Infect Dis J. 2017 Jul;36(7):645-653. doi: 10.1097/INF.0000000000001538. PMID: 28060045; PMCID: PMC5468484.

Cojutti PG, Tedeschi S, Gatti M, Zamparini E, Meschiari M, Siega PD, Mazzitelli M, Soavi L, Binazzi R, Erne EM, Rizzi M, Cattelan AM, Tascini C, Mussini C, Viale P, Pea F. Population Pharmacokinetic and Pharmacodynamic Analysis of Dalbavancin for Long-Term Treatment of Subacute and/or Chronic Infectious Diseases: The Major Role of Therapeutic Drug Monitoring. Antibiotics (Basel). 2022 Jul 24;11(8):996. doi: 10.3390/antibiotics11080996. PMID: 35892386; PMCID: PMC9331863.

2)       Parameter distribution in the study population

Authors only report means with maximum and minimum values without giving any other insight into the actual distribution of these values in the studied population. This is especially relevant for the only included covariate of body weight given the extrapolation used for the simulation.

A: Despite many other Authors report continuous variables as “Mean (Min-Max)” values, the Authors concurred with the Reviewer and amended Table 1 reporting “Median (Min-Max)” values instead of “Mean (Min-Max)” values to better define covariates distribution.

 3)       Extrapolating into unjustified territory

Given the lacking reporting on value distribution in the population, it is difficult to make authoritative statements, but the expected number of patients over 120 kg in the studied sample is in low single digits, making the extrapolation to 200 kg in the Monte Carlo simulation unjustified. See discussion on using TBW further. 

A: The Authors feel to disagree with the Reviewer on this claim, since overweight patients are well reported in the analysed population. Moreover, the Authors believe that what the Reviewer underlined is the whole meaning of using advanced mathematical techniques (i.e. nonlinear mixed-effects modelling) to describe biological processes through modelling and simulation. Having characterized the pharmacokinetic (PK) parameters of dalbavancin through a thoroughly robust estimation process (RSE <30 %, ETA shrinkage < 35%, bootstrap analysis), the same PK parameters can and should be used to simulate new scenarios to better understand how the covariate effect of total body weight may affect dalbavancin disposition.

 4)       Discussion of body weight

There is no discussion of using alternative body weight formulas such as adjBW or IBW for modelling as covariates. Given only moderate lipid solubility of dalbavancin, this could be relevant for dosing at extreme end of the distribution. While most studies on DAL use TBW as a covariate, this still needs to be discussed and explored since the paper ventures into the upper extreme of body weights where the potential bias of TBW might manifest. Given the off-label nature of these recommendations, any suggestion for dosing needs to be properly justified to avoid exposing the patient to excessive concentrations.

A: The Authors during early exploratory analysis phase and modelling steps (data non shown) explored the effect of free fat mass (FFM) formulas, given the moderate lipid solubility of dalbavancin. However, this alternative body weight measure was not found suitable to be brought forward into later modelling steps over TBW. As recognized by the Reviewer most studies on DAL use indeed TBW as a covariate, and since the whole scope of our work was to facilitate the daily work of clinicians in treating better patients the use of other body weight formulas over TBW, however mathematically stimulating, is unwarranted. In addition, if TDM is carried out drug levels are monitored and correct timing of infusion would be applied.

 5)       Excluded outliers

Please include a more thorough discussion on the nature and justification of the excluded outlier values. Just saying that their removal improved the model fit is not sufficient. 

A: The Authors feel to disagree with the Reviewer on this statement.
An outlier is defined as an aberrant observation (observed concentration) that significantly deviates from the rest of observations in a particular individual and does not refer to a subject as an outlier.

The Food and Drug administration in 1999 clearly stated that “The proportion of outliers in a dataset should be low and such points may be excluded from the analysis given the potential for these observations to negatively impact the convergence and/or parameter estimates (i.e., which may cause a bias)”

Also, Mould, D. R., & Upton, R. N. (2013) in their “Basic concepts in population modelling, simulation, and model-based drug development-part 2: introduction to pharmacokinetic modelling methods. CPT: pharmacometrics & systems pharmacology, 2(4), e38. https://doi.org/10.1038/psp.2013.14” suggest that “During data cleaning and initial model evaluations, data records may be identified as erroneous (e.g., a sudden, transient decrease in concentration) and can be commented out if they can be justified as an outlier or error that impairs model development”.

Following the latter we proceeded to comment out outliers that we identified with a CWRES >3 as aberrant (i.e. also non consistent with the concentration vs time profile of the individual, possibly due to sampling time recording error or drug quantitation error etc.)

 6)       Covariates in time

It is not clear from the text if covariates such as creatinine clearance or CRP were measured at repeated intervals (at sampling times, at repeated dosing) and if so, if the changing values were incorporated when building the model. Given the longer duration of follow-up (at least 7 days between doses, one patient with 10 doses) and the wide range of initial values, it would be interesting to see a discussion on this topic. 

A: Covariates such as creatinine or CRP were measured multiple times for patients needing blood tests at the convenience of the prescribing physician as specified in Material and methods section, paragraph 4.1, lines 358-363. Then, as per standard modelling guidance when dealing with a long-term PK study, they were treated as time varying covariates. As the reviewer is surely familiar with, NONMEM software handles time varying covariates automatically, providing the dataset is constructed correctly (i.e. EVID=2 etc.) otherwise it won’t’ even run the model.

 7)       Exploring other PK/PD parameters

Given the trough levels achieved, it seems prudent to assume the simulated profiles would also satisfy the AUC:MIC PK/PD target. Even though the authors opted to use fT<4×MIC and justified their choice, it seems that a comment on how these profiles would fare in the other parameter would be interesting. 

A: The Authors feel they have justified sufficiently their choice of PK/PD parameter of interest through a thorough literature search and historical reconstruction of events (never before reported), and discussed it concisely but otherwise completely in the Discussion section.

 8)       Missing references

There are several relevant papers that are missing from the discussion, namely a real world analysis of achieved levels in Hervochon et al. Journal of Antimicrobial Chemotherapy, Volume 78, Issue 12, December 2023, Pages 2919–2925, https://doi.org/10.1093/jac/dkad331 and more importantly analysis of protein binding in Turner   NA, Xu   A, Zaharoff   S  et al.   Determination of plasma protein binding of dalbavancin. J Antimicrob Chemother  2022; 77: 1899–902. https://doi.org/10.1093/jac/dkac131

A: The Authors added the suggested missing reference by Hervochon et al. to the Introduction Section, since it was found relevant, and the Authors wish to thank the Reviewer for the suggestion.

 However, the Authors do not believe that the addition of the reference by Tuner et al. may benefit the readability of the present manuscript or the clinical practice of dalbavancin long-term treatment. Despite being well conducted and of analytical importance the study by Turner et al. refers to in vitro data which may not reflect in vivo true protein binding of dalbavancin. Moreover, the reported 99% protein binding is in clear contrast with the label approved by FDA and EMA (namely 93%), who surely possess more thorough industry data.

However, a supposedly in vivo human protein binding of 99% would mean to maintain total drug concentration of 100 mg/L over time (see calculations below)

If  and which is clinically unthinkable, considering that many a study reported Cmax around 300 mg/L and Ctrough around 10-20 mg/L (the concentration target select by early pivotal dose selection studies to cover at least 2 weeks of treatment with one single dose). Maintaining a target of total dalbavancin > 100 mg/L would mean to re-dose patients every 3-4 days and to expose patients to relevant toxicity issue not balanced with efficacy which is achieved in real life clinical practice with repeated dosed every 3 to 4 weeks.

 9)       Covariates and Monte Carlo simulation

It is not clear what is meant on line 366 by “..so that the relationship between each covariate would be physiologically based.” when the only covariate included in the model was body weight. The paper does not mention which covariates were evaluated, only that CLCr did not improve the fit. What were the other covariates considered? One could expect albumin levels to affect free fraction and thus clearance, if there was variability in the population (values not reported by authors). 

 A: The Authors hope to clarify that when dealing with Monte Carlo Simulation one approach would be to “simply simulate” with a resampling of the same dataset used for estimation purposes (i.e. an approach typically undertaken within Monolix + Simulix softwares) or to generate a virtual dataset of “digital twins” from a virtual distribution of data generated in R, or even better still by harvesting covariate data from real published dataset (an approach typically used from advanced NONMEM users). Our case is the latter. We harvested necessary covariate data from real CDC governmental websites (as reported in the methods) in order to maintain a physiological correlation between the harvested covariates data (ie age, weight, height etc), a physiological correlation which when generated in R or resampled from original dataset (i.e. with Simulix) would be lost. However, we only used only TBW for simulation purposes, but only because any other covariate was not brought forward in the final model.

The Authors fear there is some misunderstanding on modelling estimation phase and the simulation phase that followed the former due to layout formatting rules imposed by the Journal. The modelling evaluation steps (with covariate search methods) are provided at the end of the manuscript in the Material and Method’s Section, paragraph 4.2, lines 378-388.

10)   Minor issues

A minor quibble, but in Figure 1, the axes labels are inconsistent - X axes should also include units for completeness and Y axes should make it clear that those are measured levels (as technically, both axes are dalbavalcin). This may be clear to people familiar with goodness of fit plots, but for completeness the labels should be correct. 

A: The Authors updated Figure 1 as kindly suggested by the Reviewer.

On line 95, please consider mentioning that the cited EUCAST clinical breakpoint cited is no longer current, as the paper itself uses the new updated breakpoint of 0.25 mg/L. 

A: The Authors wish to thank the Reviewer for noticing EUCAST update of dalbavancin clinical breakpoint on the 1st of January 2025. The Authors anticipated this update would be due soon, thus during the previous months of pharmacokinetic analysis proceeded to set the breakpoint already to 0.25 mg/L as suggested by USCAST. Following the reviewer suggestion manuscript text and bibliography have been updated.

 Despite the study not being designed or powered to assess outcomes, it would be interesting to see the real world MIC values in the pathogens that were identified and treatment outcomes, if available. 

A: The Authors thank the Reviewer for highlighting further scope of research. Unfortunately, dalbavancin MICs measurements are not available in hospital worldwide since dalbavancin MICs must be determined in the presence of polysorbate-80 in the medium for broth dilution methods and agar dilution methods have not yet been validated.

Reviewer 2 Report

Comments and Suggestions for Authors

The reviewed article presents an evaluation of the antibiotic drug Dalbavancin, administered for the treatment of Gram-positive bacterial infections, studied through Pharmacokinetic-Pharmacodynamic modeling.
This type of study requires an appropriate statistical analysis, and the authors have presented a highly rigorous methodology. They utilize two specialized software tools, NONMEM and R, with very specific applications within the study. I found this approach to be highly appropriate and executed in an excellent manner.

The introduction provides sufficient context to understand the issues addressed in the research study.
The study was conducted with a human population, and the results were handled appropriately.
The conclusions are consistent with the findings obtained during the modeling process, and I do not observe any discrepancies in their interpretation. It is noteworthy to leverage the pharmacokinetic properties of Dalbavancin to modify its traditional administration regimens. In this case, the authors recommend making personalized adjustments for each patient based on body weight. The results presented are highly appropriate given the doses tested in the described administration scheme.

There is limited information on this type of application schemes, which gives the article significant scientific relevance.
The presented graphs are descriptive and of high quality, effectively highlighting the described results.
No instances of plagiarism were found, demonstrating that this text is original and based on the extensive experience of the authors.
The article includes documentation of ethical committee approval, as required for this type of study, which is mandatory.
No inappropriate self-citations were identified.
The article meets the editorial requirements, and in my opinion, no further revisions are necessary for publication.

Author Response

The reviewed article presents an evaluation of the antibiotic drug Dalbavancin, administered for the treatment of Gram-positive bacterial infections, studied through Pharmacokinetic-Pharmacodynamic modeling.
This type of study requires an appropriate statistical analysis, and the authors have presented a highly rigorous methodology. They utilize two specialized software tools, NONMEM and R, with very specific applications within the study. I found this approach to be highly appropriate and executed in an excellent manner.

The introduction provides sufficient context to understand the issues addressed in the research study.
The study was conducted with a human population, and the results were handled appropriately.
The conclusions are consistent with the findings obtained during the modeling process, and I do not observe any discrepancies in their interpretation. It is noteworthy to leverage the pharmacokinetic properties of Dalbavancin to modify its traditional administration regimens. In this case, the authors recommend making personalized adjustments for each patient based on body weight. The results presented are highly appropriate given the doses tested in the described administration scheme.

There is limited information on this type of application schemes, which gives the article significant scientific relevance.
The presented graphs are descriptive and of high quality, effectively highlighting the described results.
No instances of plagiarism were found, demonstrating that this text is original and based on the extensive experience of the authors.
The article includes documentation of ethical committee approval, as required for this type of study, which is mandatory.
No inappropriate self-citations were identified.
The article meets the editorial requirements, and in my opinion, no further revisions are necessary for publication.

The Authors wish to sincerely thank the Reviewer for appreciating their hard work developed through a time-consuming sampling phase, month of pharmacokinetic analysis, and thorough literature synthesis. The recognition of the hard work done by peers is what encourages in still doing research for the improvement of the clinical care of patients.

Reviewer 3 Report

Comments and Suggestions for Authors

The article presents a population pharmacokinetic analysis of DAL, a long-acting lipoglycopeptide antibiotics, and simulates the optimal multidose regimen for DAL that achieves the PK/PD target of 100%fT>4×MIC in patients requiring long-term treatment. Overall the manuscript is well-organized and meaningful, but I have several comments and suggestions. 

1. Given that the population PK analysis includes patients with 8 different sites of infection, and the authors simulated the difficult-to-treat infection conditions, has the site of infection been tested as a categorical covariate in the popPK model to show site of infection/indication does not affect the PK of DAL?

2. The total drug concentration was measured and characterized in the population PK model. Has free drug concentration ever been measured or was the free fraction of 7% fixed in the simulation to calculate the free-drug AUC for all patients? The reason I ask is that protein binding fraction is often altered in the renal impairment patients. The CrCL in the popPK studies ranges from 31.7 to 283 mL/min which includes the mild renal impairment patients and moderate renal impairment patients. Does the 93% protein binding still apply in those renal impairment population and could any changes in protein binding affect the simulation results?

3. Consider adding a paragraph at the end of the introduction to clearly state the objective and purpose of this study.

4. Has any other base model been evaluated, or was the two-compartment model with first-order elimination chosen based on prior experience? Consider provide the rationale or criteria for selecting the current base model.

5. Body weight was identified as a significant covariate on PK parameters in the final popPK model. Please include the estimated values and %RSE for the exponent values of body weight on the PK parameters to better demonstrate its covariate effect.

6. The demographic data shows a body weight range of 44–179 kg in the population PK analysis, while simulations include a broader range (40–200 kg). Could you please clarify the rationale and potential issues with a broader demographic range in the simulation?

7. Just curious, did the authors perform a sensitivity analysis, or if not, based on your understanding of the data, which specific parameters may the optimal dosing schedule be sensitive to?

8. Multiple popPK analyses have been previously performed for dalbavancin PK. Besides the one cited in reference 23, which had different popPK results as this work, do other cited studies (Reference 6, 23, 24, 25, 26) have consistent results as this analysis?

9. Table 1: Seems missing the "1" value for S.lugdunensis. Otherwise the sum of Isolate is 29 instead of 30.

10. Consider providing detailed captions for Figures 1, 3, and 4 to improve clarity and understanding of the data presented.

11. The title uses inconsistent capitalization. Please revise the title to ensure consistent use of first-letter uppercase for each word.

12. What is TDM? Therapeutic drug monitoring? Please provide the full term for this abbreviation when first mentioned in the article.

Author Response

The article presents a population pharmacokinetic analysis of DAL, a long-acting lipoglycopeptide antibiotics, and simulates the optimal multidose regimen for DAL that achieves the PK/PD target of 100%fT>4×MIC in patients requiring long-term treatment. Overall the manuscript is well-organized and meaningful, but I have several comments and suggestions. 

A: The Authors wish to thank the Reviewer for his/her suggestions in bringing improvements to our manuscript.

  1. Given that the population PK analysis includes patients with 8 different sites of infection, and the authors simulated the difficult-to-treat infection conditions, has the site of infection been tested as a categorical covariate in the popPK model to show site of infection/indication does not affect the PK of DAL?

A: The Authors tested the site of infection as categorical covariate against the pharmacokinetic parameters during the Exploratory Analysis phase. However, it wasn’t found significant to be brought forward into the modelling phase. This may be explained by the highly tissue distributive proprieties of dalbavancin, so that site of infection should not affect dalbavancin disposition.
As specified in lines 389-393, dalbavancin free fraction in plasma (7%) is expected to be freely available for antimicrobial activity even to poorly accessible sites of infections.

  1. The total drug concentration was measured and characterized in the population PK model. Has free drug concentration ever been measured or was the free fraction of 7% fixed in the simulation to calculate the free-drug AUC for all patients? The reason I ask is that protein binding fraction is often altered in the renal impairment patients. The CrCL in the popPK studies ranges from 31.7 to 283 mL/min which includes the mild renal impairment patients and moderate renal impairment patients. Does the 93% protein binding still apply in those renal impairment population and could any changes in protein binding affect the simulation results?

A: We thank the referee for his/her comment about the possible influence of renal impairment on the plasma protein binding of the drug, a topic that is widely discussed in the literature.

Keller F, Maiga M, Neumayer HH, Lode H, Distler A. Pharmacokinetic effects of altered plasma protein binding of drugs in renal disease. Eur J Drug Metab Pharmacokinet. 1984;9(3):275-82;

Celestin MN, Musteata FM. Impact of Changes in Free Concentrations and Drug-Protein Binding on Drug Dosing Regimens in Special Populations and Disease States. J Pharma Sci. 2021;110(10):3331-3344.

However, we have considered that plasma protein binding of dalbavancin (93%) is not influenced by drug concentrations, renal and liver impairment, as stressed by the dalbavancin technical note. Therefore, we adopted a free fraction of dalbavancin corresponding to 7% for our modelling and simulation.

The free fraction of 7% chosen for simulation purposes was based on label data of dalbavancin from Regulatory Agencies (see SmPC Xydalba in References n. 2), moreover measuring dalbavancin total drug concentration is recommended over the free fraction itself (as for many other drugs for which Therapeutic Drug Monitoring is warranted), in addition the in vitro measurement of free drug concentration does not fully reflect the in vivo situation. Changes in the percentage of free drug/bound to plasma proteins are usually gradual and do not alter the effects of the drug, because the compensatory increase in the rate of elimination (Clearance) from the body prevents the accumulation of a high concentration of bound drug in the plasma. Therefore, the degree of protein binding is probably negligible in patients with mild renal failure, and in any case, the renal function we evaluated as a covariate did not affect the simulation results.

  1. Consider adding a paragraph at the end of the introduction to clearly state the objective and purpose of this study.

A: A sentence has now been added in the Introduction, as suggested by the Reviewer.

  1. Has any other base model been evaluated, or was the two-compartment model with first-order elimination chosen based on prior experience? Consider provide the rationale or criteria for selecting the current base model.

A: Due to layout formatting rules imposed by the Journal, the modelling evaluation steps are provided at the end of the manuscript in the Material and Method’s Section, paragraph 4.2, lines 346-348.

  1. Body weight was identified as a significant covariate on PK parameters in the final popPK model. Please include the estimated values and %RSE for the exponent values of body weight on the PK parameters to better demonstrate its covariate effect.

A: As specified in lines 132-133 body weight scaled by allometric principles with fixed exponents [20–22] was set a priori as a covariate on clearances (CL, Q) and volumes of distribution (V1, V2). However, as suggested, the Authors improved the sentence readability specifying exponent valued (though fixed by allometric principles, citations 20-22).

  1. The demographic data shows a body weight range of 44–179 kg in the population PK analysis, while simulations include a broader range (40–200 kg). Could you please clarify the rationale and potential issues with a broader demographic range in the simulation?

A: The Authors believe that what the Reviewer underlined is the whole meaning of using advanced mathematical techniques (i.e. nonlinear mixed-effects modelling) to describe biological processes. Having characterized the pharmacokinetic (PK) parameters of dalbavancin through a thoroughly robust estimation process, the same PK parameters can and should be used to simulate new scenarios to better understand how the covariate effect of total body weight may affect dalbavancin disposition.

  1. Just curious, did the authors perform a sensitivity analysis, or if not, based on your understanding of the data, which specific parameters may the optimal dosing schedule be sensitive to?

A: The Authors performed the standard Exploratory Data Analysis evaluating time vs concentration data to select the best compartmental model, parameters vs covariate analysis to select the covariate model etc.

  1. Multiple popPK analyses have been previously performed for dalbavancin PK. Besides the one cited in reference 23, which had different popPK results as this work, do other cited studies (Reference 6, 23, 24, 25, 26) have consistent results as this analysis?

A: Yes, the other studies present results consistent with our own. However, in some aspects our study is more robust than some of the others cited studies, since it presents a rigorous estimation of the pharmacokinetic parameters (RSE <30%, Eta shrinkage < 35%, bootstrap analysis) that many of the others studies fail to possess in some parameters.

  1. Table 1: Seems missing the "1" value for S.lugdunensis. Otherwise the sum of Isolate is 29 instead of 30.

A: The Authors thank the Reviewer for noticing the missing value. Table 1 has been revised accordingly.

  1. Consider providing detailed captions for Figures 1, 3, and 4 to improve clarity and understanding of the data presented.

A: The Authors improved Figure 1, and descriptions of Figure 3 and 4 are presented in the Results section of the manuscript

  1. The title uses inconsistent capitalization. Please revise the title to ensure consistent use of first-letter uppercase for each word.

A: The Authors revised the title accordingly.

  1. What is TDM? Therapeutic drug monitoring? Please provide the full term for this abbreviation when first mentioned in the article.

A: The Authors added TDM definition in line 112 as suggested.

Round 2

Reviewer 1 Report

Comments and Suggestions for Authors

I wish to thank the authors for their thoughtful and constructive replies. I do agree with most of their arguments and as I stated in my review, most "objections" were aimed at better presentation and discussion of the topics raised. The point of NONMEM handling time varying covariates automatically does not - in my opinion - negate the need to include the data or at least comment on them in the manuscript. As it is, it was not clear how many of the patients had updated data and at what intervals (regardless of this being left to the treating doctors). 

I understand the authors' trust in extrapolating their data but still find it unwarranted. As mentioned, we are deep into off-label territory here and I would very much be hesitant to base my clinical decisions on a population model based on such assumptions. Studies on antibiotic kinetics in the extremely obese have shown repeatedly that extrapolation is not automatically guaranteed. 

Reviewer 3 Report

Comments and Suggestions for Authors

No additional comments